# State Aware Imitation Learning

**Yannick Schroecker**
College of Computing
Georgia Institute of Technology
yannickschroecker@gatech.edu

**Charles Isbell**
College of Computing
Georgia Institute of Technology
isbell@cc.gatech.edu

## Abstract

Imitation learning is the study of learning how to act given a set of demonstrations provided by a human expert. It is intuitively apparent that learning to take optimal actions is a simpler undertaking in situations that are similar to the ones shown by the teacher. However, imitation learning approaches do not tend to use this insight directly. In this paper, we introduce State Aware Imitation Learning (SAIL), an imitation learning algorithm that allows an agent to learn how to remain in states where it can confidently take the correct action and how to recover if it is lead astray. Key to this algorithm is a gradient learned using a temporal difference update rule which leads the agent to prefer states similar to the demonstrated states. We show that estimating a linear approximation of this gradient yields similar theoretical guarantees to online temporal difference learning approaches and empirically show that SAIL can effectively be used for imitation learning in continuous domains with non-linear function approximators used for both the policy representation and the gradient estimate.

## 1 Introduction

One of the foremost challenges in the field of Artificial Intelligence is to program or train an agent to act intelligently without perfect information and in arbitrary environments. Many avenues have been explored to derive such agents but one of the most successful and practical approaches has been to learn how to imitate demonstrations provided by a human teacher. Such imitation learning approaches provide a natural way for a human expert to program agents and are often combined with other approaches such as reinforcement learning to narrow the search space and to help find a near optimal solution. Success stories are numerous in the field of robotics [3] where imitation learning has long been subject of research but can also be found in software domains with recent success stories including AlphaGo [23] which learns to play the game of Go from a database of expert games before improving further and the benchmark domain of Atari games where imitation learning combined with reinforcement learning has been shown to significantly improve performance over pure reinforcement learning approaches [9].

Formally, we define the problem domain as a Markov decision process, i.e. by its states, actions and unknown Markovian transition probabilities $p(s'|s, a)$ of taking action $a$ in state $s$ leading to state $s'$. Imitation learning aims to find a policy $\pi(a|s)$ that dictates the action an agent should take in any state by learning from a set of demonstrated states $S_D$ and the corresponding demonstrated actions $A_D$. The likely most straight-forward approach to imitation learning is to employ a supervised learning algorithm such as neural networks in order to derive a policy, treating the demonstrated states and actions as training inputs and outputs respectively. However, while this can work well in practice and has a long history of successes starting with, among other examples, early ventures into autonomous driving[18], it also violates a key assumption of statistical supervised learning by having past predictions affect the distribution of inputs seen in the future. It has been shown that agents trained this way have a tendency to take actions that lead it to states that are dissimilar from

any encountered during training and in which the agent is less likely to have an accurate model of how to act [18, 19]. Deviations from the demonstrations based on limitations of the learning model or randomness in the domain are therefore amplified as time progresses. Several approaches exist that are capable of addressing this problem. Interactive imitation learning methods (e.g. [5, 19, 20]) address this problem directly but require continuing queries to the human teacher which is often not practical. Inverse Reinforcement Learning (IRL) approaches attempt to learn the objective function that the demonstrations are optimizing and show better generalization capabilities. However, IRL approaches often require a model of the domain, can be limited by the representation of the reward function and are learning a policy indirectly. A consequence of the latter is that small changes to the learned objective function can lead to large changes in the learned policy.

In this paper we introduce State Aware Imitation Learning (SAIL). SAIL aims to address the aforementioned problem by explicitly learning to reproduce demonstrated trajectories based on their states as well as their actions. Intuitively, if an agent trained with SAIL finds itself in a state similar to a demonstrated state it will prefer actions that are similar to the demonstrated action but it will also prefer to remain near demonstrated states where the trained policy is more likely to be accurate. An agent trained with SAIL will thus learn how to recover if it deviates from the demonstrated trajectories. We achieve this in a principled way by finding the maximum-a-posteriori (MAP) estimate of the complete trajectory. Thus, our objective is to find a policy which we define to be a parametric distribution $\pi_\theta(a|s)$ using parameters $\theta$. Natural choices would be linear functions or neural networks. The MAP problem is then given by

$$\text{argmax}_\theta p(\theta|S_D, A_D) = \text{argmax}_\theta \log p(A_D|S_D, \theta) + \log p(S_D|\theta) + \log p(\theta). \qquad (1)$$

Note that this equation differs from the naive supervised approach in which the second term $\log p(S_D|\theta)$ is assumed to be independent from the current policy and is thus irrelevant to the optimization problem. Maximizing this term leads to the agent actively trying to reproduce states that are similar to the ones in $S_D$. It seems natural that additional information about the domain is necessary in order to learn how to reach these states. In this work, we obtain this information using unsupervised interactions with the environment. We would like to stress that our approach does not require further input from the human teacher, any additional measure of optimality, or any model of the environment. A key component of our algorithm is based on the work of Morimura et al.[15] who estimate a gradient of the distribution of states observed when following the current policy using a least squares temporal difference learning approach and use their results to derive an alternative policy gradient algorithm. We discuss their approach in detail in section 3.1 and extend the idea to an online temporal difference learning approach in section 3.2. This adaptation gives us greater flexibility for our choice of function approximator and also provides a natural way to deal with an additional constraint to the optimization problem which we will introduce below. In section 3.3, we describe the full SAIL algorithm in detail and show that the estimated gradient can be used to derive a principled and novel imitation learning approach. We then evaluate our approach on a tabular domain in section 4.1, comparing our results to a purely supervised approach to imitation learning as well as to sample based inverse reinforcement learning. In section 4.2 we show that SAIL can successfully be applied to learn a neural network policy in a continuous bipedal walker domain and achieves significant improvements over supervised imitation learning in this domain.

## 2   Related works

One of the main problems SAIL is trying to address is the problem of remaining close to states where the agent can act with high confidence. We identify three different classes of imitation learning algorithms that address this problem either directly or indirectly under different assumptions and with different limitations. A specialized solution to this problem can be found in the field of robotics. Imitation learning approaches in robotics often do not aim to learn a full policy using general function approximators but instead try to predict a trajectory that the robot should follow. Trajectory representations such as Dynamic Movement Primitives [21] give the robot a sequence of states (or its derivatives) which the robot then follows using a given control law. The role of the control law is to drive the robot towards the demonstrated states which is also a key objective of SAIL. However, this solution is highly domain specific and a controller needs to be chosen that fits the task and representation of the state space. It can, for example, be more challenging to use image based state representations. For a survey of imitation learning methods applied to robotics, see [3].

The second class of algorithms is what we will call iterative imitation learning algorithms. A key characteristic of these algorithms is that the agent actively queries the expert for demonstrations in states that it sees when executing its current policy. One of the first approaches in this class is SEARN[5]. When applied to Imiteration Learning, SEARN starts by following the experts action at every step, then iteratively uses the demonstrations collected during the last episode to train a new policy and collects new episodes by taking actions according to a mixture of all previously trained policies and the experts actions. Over time SEARN learns to follow its mixture of policies and stops relying on the expert to decide which actions to take. Ross et al. [19] first proved that the pure supervised approach to imitation learning can lead to the error rate growing over time. To alleviate this issue they introduced a similar iterative algorithm called SMILe and proved that the error rate increases near linearly with respect to the time horizon. Building on this, Ross et al. introduced DAGGER [20]. DAGGER provides similar theoretical guarantees and empirically outperforms SMILe by augmenting a single training set during each iteration based on queries to the expert on the states seen during execution. DAGGER does not require previous policies to be stored in order to calculate a mixture. Note that while these algorithms are guaranteed to address the issue of straying too far from demonstrations, they approach the problem from a different direction. Instead of preferring states on which the agent has demonstrations, the algorithms collects more demonstrations in states the agent actually sees during execution. This can be effective but requires additional interaction with the human teacher which is often not cheaply available in practice.

As mentioned above, our approach also shares significant similarities with Inverse Reinforcement Learning (IRL) approaches [17]. IRL methods aim to derive a reward function for which the provided demonstrations are optimal. This reward function can then be used to compute a complete policy. Note that the IRL problem is known to be ill-formed as a set of demonstrations can have an infinite amount of corresponding reward functions. Successful approaches such as Maximum Entropy IRL (MaxEntIRL) [27] thus attempt to disambiguate between possible reward functions by reasoning explicitly about the distribution of both states and actions. In fact, Choi and Kim [4] argue that many existing IRL methods can be rewritten as finding the MAP estimate for the reward function given the provided demonstrations using different probabilistic models. This provides a direct link to our work which maximizes the same objective but with respect to the policy as opposed to the reward function. A significant downside of many IRL approaches is that they require a model describing the dynamics of the world. However, sample based approaches exist. Boularias et al. [1] formulate an objective function similar to MaxEntIRL but find the optimal solution based on samples. Relative Entropy IRL (RelEntIRL) aims to find a reward function corresponding to a distribution over trajectories that matches the observed features while remaining within a relative entropy bound to the uniform distribution. While RelEntIRL can be effective, it is limited to linear reward functions. Few sample based methods exist that are able to learn non-linear reward functions. Recently, Finn et al. proposed Guided Cost Learning [6] which optimizes an objective based on MaxEntIRL using importance sampling and iterative refinement of the sample policy. Refinement is based on optimal control with learned models and is thus best suited for problems in domains in which such methods have been shown to work well, e.g. robotic manipulation tasks. A different direction for sample based IRL has been proposed by Klein et al. who treat the scores of a score-based classifier trained using the provided demonstration as a value function, i.e. the long-term expected reward, and use these values to derive a reward function. Structured Classification for IRL (SCIRL) [13] uses estimated feature expectations and linearity of the value function to derive the parameters of a linear reward function while the more recent Cascaded Supervised IRL (CSI) [14] derives the reward function by training a Support Vector Machine based on the observed temporal differences. While non-linear classifiers could be used, the method is dependent on the interpretability of the score as a value function. Recently, Ho et al.[11] introduced an approach that aims to find a policy that implicitly maximizes a linear reward function but without the need to explicitly represent such a reward function. Generative Adversarial Imitation Learning [10] uses a method similar to Generative Adversarial Networks[7] to extend this approach to nonlinear reward functions. The resulting algorithm trains a discriminator to distinguish between demonstration and sampled trajectory and uses the probability given by the discriminator as a reward to train a policy using reinforcement learning. The maximum likelihood approach presented here can be seen as an approximation of minimizing the KL divergence between the demonstrated states and actions and the reproduction by the learned policy. This can also be achieved by using the ratio of state-action probabilities $\frac{p_D(a,s)}{d^{\pi_\theta}(s)\pi_\theta(a|s)}$ as a reward which is a straight-forward transformation of the output of the optimal discriminator[7]. Note however that this equality only holds assuming an infinite number of demonstrations. Furthermore note that unlike the

gradient network introduced in this paper, the discriminator needs to learn about the distribution of the expert's demonstrations.

Finally, we would like to point out the similarities our work shares with meta learning techniques that learn the gradients (e.g.[12]) or determine the weight updates (e.g. [22], [8]) for a neural network. Similar to these meta learning approaches, we propose to estimate the gradient w.r.t. the policy. While a complete review of this work is beyond the scope of this paper, we believe that many of the techniques developed to address challenges in this field can be applicable to our work as well.

## 3   Approach

SAIL is a gradient ascent based algorithm to finding the true MAP estimate of the policy. A significant role in estimating the gradient $\nabla_\theta \log p(\theta|S_D, A_D)$ will be to estimate the gradient of the (stationary) state distribution induced by following the current policy. We write the stationary state distribution as $d^{\pi_\theta}(s)$, assume that the Markov chain is ergodic (i.e. the distribution exists) and review the work by Morimura et al. [15] on estimating its gradient $\nabla_\theta \log d^{\pi_\theta}(s)$ in section 3.1. We outline our own online adaptation to retrieve this estimate in section 3.2 and use it in order to derive the full SAIL gradient $\nabla_\theta \log p(\theta|S_D, A_D)$ in section 3.3.

### 3.1   A temporal difference approach to estimating $\nabla_\theta \log d^\pi(s)$

We first review the work by Morimura et al. [15] who first discovered a relationship between the gradient $\nabla_\theta \log d^{\pi_\theta}(s)$ and value functions as used in the field of reinforcement learning. Morimura et al. showed that the gradient can be written recursively and decomposed into an infinite sum so that a corresponding temporal difference loss can be derived.

By definition, the gradient of the stationary state distribution in a state $s'$ can be written in terms of prior states $s$ and actions $a$.

$$\nabla_\theta d^{\pi_\theta}(s') = \nabla_\theta \int d^{\pi_\theta}(s)\pi_\theta(a|s)p(s'|s,a)ds, a \tag{2}$$

Using $\nabla_\theta(d^{\pi_\theta}(s)\pi_\theta(a|s)p(s'|s,a)) = p(s,a,s')(\nabla_\theta \log d^{\pi_\theta}(s) + \nabla_\theta \log \pi_\theta(a|s))$ and dividing by $d^{\pi_\theta}(s')$ on both sides, we obtain

$$0 = \int q(s,a|s')\left(\nabla_\theta \log d^{\pi_\theta}(s) + \nabla_\theta \log \pi_\theta(a|s) - \nabla_\theta \log d^{\pi_\theta}(s')\right)ds, a \tag{3}$$

Where q denotes the reverse transition probabilities. This can be seen as an expected temporal difference error over the previous state and action where the temporal difference error is defined as

$$\delta(s,a,s') := \nabla_\theta \log d^{\pi_\theta}(s) + \nabla_\theta \log \pi_\theta(a|s) - \nabla_\theta \log d^{\pi_\theta}(s') \tag{4}$$

In the original work, Morimura et al. derive a least squares estimator for $\nabla_\theta \log d^{\pi_\theta}(s')$ based on minimizing the expected squared temporal difference error as well as a penalty to enforce the constraint $E[\nabla_\theta \log d^{\pi_\theta}(s)] = 0$, ensuring $d^{\pi_\theta}$ remains a proper probability distribution, and apply it to policy gradient reinforcement learning. In the following sections we formulate an online update rule to estimate the gradient, argue convergence in the linear case, and use the estimated gradient to derive a novel imitation learning algorithm.

### 3.2   Online temporal difference learning for $\nabla_\theta \log d^\pi(s)$

In this subsection we define the online temporal difference update rule for SAIL and show that convergence properties are similar to the case of average reward temporal difference learning[25]. Online temporal difference learning algorithms are computationally more efficient than their least squares batch counter parts and are essential when using high-dimensional non-linear function approximations to represent the gradient. We furthermore show that online methods give us a natural way to enforce the constraint $E[\nabla_\theta \log d^{\pi_\theta}(s)] = 0$. We aim to approximate $\nabla_\theta \log d^\pi(s)$ up to an unknown constant vector $c$ and thus define our target as $f^*(s) := \nabla_\theta \log d^\pi(s) + c$. We use a temporal difference update to learn a parametric approximation $f_\omega(s) \approx f^*(s)$. The update rule based on taking action $a$ in state $s$ and transitioning to state $s'$ is given by

$$\omega_{k+1} = \omega_k + \alpha \nabla_\omega f_\omega(s')\left(f_\omega(s) + \nabla_\theta \log \pi(a|s) - f_\omega(s')\right). \tag{5}$$

---

**Algorithm 1** State Aware Imitation Learning

---
1: **function** SAIL($\omega, \alpha_\theta, \alpha_\omega, S_D, A_D$)
2:    $\theta \leftarrow$ SupervisedTraining($S_D, A_D$)
3:    **for** $k \leftarrow 0..\#$Iterations **do**
4:      $S_E, A_E \leftarrow$ CollectUnsupervisedEpisode($\pi_\theta$))
5:      $\omega \leftarrow \omega + \alpha_\omega \frac{1}{|S_E|} \sum_{s,a,s' \in \text{transitions}(S_E, A_E)} (f_\omega(s) + \nabla_\theta \log \pi_\theta(a|s) - f_\omega(s')) \nabla_\omega f(s')$
6:      $\mu \leftarrow \frac{1}{|S_E|} \sum_{s \in S_E} f_\omega(s)$
7:      $\theta \leftarrow \theta + \alpha_\theta \left( \frac{1}{|S_D|} \sum_{s,a \in \text{pairs}(S_D, A_D)} (\nabla_\theta \log \pi_\theta(a|s) + (f_\omega(s) - \mu)) + \nabla_\theta p(\theta) \right)$
      **return** $\theta$

---

Note that if $f_\omega$ converges to an approximation of $f^*$ then due to $E[\nabla_\theta \log d^{\pi_\theta}(s)] = 0$, we have $\nabla_\theta \log d^\pi(s) \approx f_\omega(s) - E[f_\omega(s)]$ where the expectation can be estimated based on samples.

While convergence of temporal difference methods is not guaranteed in the general case, some guarantees can be made in the case of linear function approximation $f_\omega(s) := \omega^T \phi(s)$[25]. We note that $E[\nabla_\theta \log \pi(a|s)] = 0$ and thus for each dimension of $\theta$ the update can be seen as a variation of average reward temporal difference learning where the scalar reward is replaced by the gradient vector $\nabla_\theta \log \pi(a|s)$ and $f_\omega$ is bootstrapped based on the previous state as opposed to the next. While the role of current and next state in this update rule are reversed and this might suggest that updates should be done in reverse, the convergence results by Tsitsiklis and Van Roy[25] are dependent only on the limiting distribution of following the sample policy on the domain which remains unchanged regardless of the ordering of updates [15]. It is therefore intuitively apparent that the convergence results still hold and that $f_\omega$ converges to an approximation of $f^*$. We formalize this notion in Appendix A.

**Introducing a discount factor** So far we related the update rule to average reward temporal difference learning as this was a natural consequence of the assumptions we were making. However, in practice we found that a formulation analogous to discounted reward temporal difference learning may work better. While this can be seen as a biased but lower variance approximation to the average reward problem [26], a perhaps more satisfying justification can be obtained by reexamining the simplifying assumption that the sampled states are distributed by the stationary state distribution $d^{\pi_\theta}$. An alternative simplifying assumption is that the previous states are distributed by a mixture of the starting state distribution $d_0(s_{-1})$ and the stationary state distribution $p(s_{-1}) = (1 - \gamma)d_0(s_{-1}) + \gamma d^\pi(s_{-1})$ for $\gamma \in [0, 1]$. In this case, equation 3 has to be altered and we have

$$0 = \int p(s, a|s') \left( \gamma \nabla_\theta \log d^{\pi_\theta}(s) + (1 - \gamma)\nabla_\theta \log d_0(s) + \nabla_\theta \log \pi_\theta(a|s) - \nabla_\theta \log d^{\pi_\theta}(s') \right) ds, a.$$

Note that $\nabla_\theta \log d_0(s) = 0$ and thus we recover the discounted update rule

$$\omega_{k+1} = \omega_k + \alpha \nabla_\omega f(s') \left( \gamma f(s) + \nabla_\theta \log \pi(a|s) - f(s') \right) \tag{6}$$

### 3.3 State aware imitation learning

Based on this estimate of $\nabla_\theta \log d^{\pi_\theta}$ we can now derive the full State Aware Imitation Learning algorithm. SAIL aims to find the full MAP estimate as defined in Equation 1 via gradient ascent. The gradient decomposes into three parts:

$$\nabla_\theta \log p(\theta|S_D, A_D) = \nabla_\theta \log p(A_D|S_D, \theta) + \nabla_\theta \log p(S_D|\theta) + \nabla_\theta \log p(\theta) \tag{7}$$

The first and last term make up the gradient used for gradient descent based supervised learning and can usually be computed analytically. To estimate $\nabla_\theta \log p(S_D|\theta)$, we disregard information about the order of states and make the simplifying assumptions that all states are drawn from the stationary distribution. Under this assumption, we can estimate $\nabla_\theta \log p(S_D|\theta) = \sum_{s \in S_D} \nabla_\theta \log d^{\pi_\theta}(s)$ based on unsupervised transition samples using the approach described in section 3.2. The full SAIL algorithm thus maintains a current policy as well an estimate of $\nabla_\theta \log p(S_D|\theta)$ and iteratively

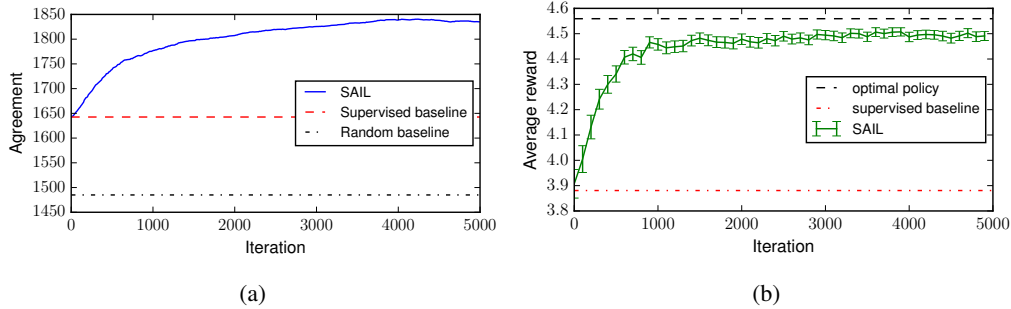

Figure 1: a) The sum of probabilities of taking the optimal action double over the baseline. b) The reward $(+/-2\sigma)$ obtained after 5000 iterations of SAIL is much closer to the optimal policy.

1. Collects unsupervised state and action samples $S_E$ and $A_E$ from the current policy,

2. Updates the gradient estimate using Equation 5 and estimates $E[f_\omega(s)]$ using the sample mean of the unsupervised states or an exponentially moving sample mean

$$\mu := \frac{1}{|S_E|} \sum_{s \in S_E} f_\omega(s)$$

3. Updates the current policy using the estimated gradient $f_\omega(s) - \mu$ as well as the analytical gradients for $\nabla_\theta \log p(\theta)$ and $\nabla_\theta \log p(A_D|S_D, \theta)$. The SAIL gradient is given by

$$\nabla_\theta \log p(\theta|S_D, A_D) = \sum_{s,a \in \text{pairs}(S_D, A_D)} (f_\omega(s) - \mu + \nabla_\theta \log p(a|s, \theta)) + \nabla_\theta p(\theta)$$

The full algorithm is also outlined in Algorithm 1.

## 4   Evaluation

We evaluate our approach on two domains. The first domain is a harder variation of the tabular racetrack domain first used in [1] with 7425 states and 5 actions. In section 4.1.1, we use this domain to show that SAIL can improve on the policy learned by a supervised baseline and learn to act in states the policy representation does not generalize to. In section 4.1.2 we evaluate sample efficiency of an off-policy variant of SAIL. The tabular representation allows us to compare the results to RelEntIRL [1] as a baseline without restrictions arising from the chosen representation of the reward function. The second domain we use is a noisy variation of the bipedal walker domain found in OpenAI gym[2]. We use this domain to evaluate the performance of SAIL on tasks with continuous state and action spaces using neural networks to represent the policy as well as the gradient estimate and compare it against the supervised baseline using the same representations.

### 4.1   Racetrack domain

We first evaluate SAIL on the racetrack domain. This domain is a more difficult variation of the domain used by Boularias et al. [1] and consists of a grid with 33 by 9 possible positions. Each position has 25 states associated with it, encoding the velocity (-2, -1, 0, +1, +2) in the x and y direction which dictates the movement of the agent at each time step. The domain has 5 possible actions allowing the agent to increase or reduce its velocity in either direction or to keep its current velocity. Randomness is introduced to the domain using the notion of a failure probability which is set to be $0.8$ if the absolute velocity in either direction is 2 and $0.1$ otherwise. The goal of the agent is to complete a lap around the track without going off-track which we define to be the area surrounding the track ($x = 0, y = 0, x > 31$ or $y > 6$) as well as the inner rectangle ($2 < x < 31$ and $2 < y < 6$). Note that unlike in [1], the agent has the ability to go off-track as opposed to being constrained by a wall and has to learn to move back on track if random chance makes it stray from it. Furthermore, the probability of going off-track is higher as the track is more narrow in this variation of the domain. This makes the domain more challenging to learn using imitation learning alone.

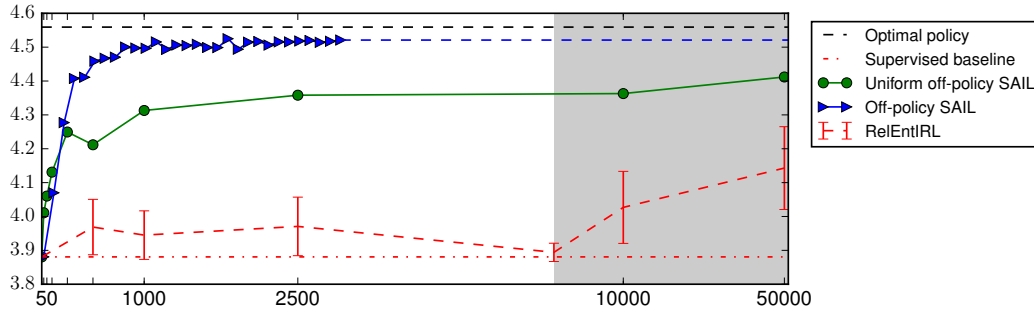

Figure 2: Reward obtained using off-policy training. SAIL learns a near-optimal policy using only 1000 sample episodes. The scale is logarithmic on the x-axis after 5000 iterations (gray area).

For all our experiments, we use a set of 100 episodes collected from an oracle. To measure performance, we assign a score of $-0.1$ to being off-track, a score of $5$ for completing the lap and $-5$ for crossing the finish line the wrong way. Note that this score is not used during training but is purely used to measure performance in this evaluation. We also use this score as a reward to derive an oracle.

### 4.1.1 On-policy results

For our first experiment, we compare SAIL against a supervised baseline. As the oracle is deterministic and the domain is tabular, this means taking the optimal action in states encountered as part of one of the demonstrated episodes and uniformly random actions otherwise. For the evaluation of SAIL, we initialize the policy to the supervised baseline and use the algorithm to improve the policy over 5000 iterations. At each iteration, 20 unsupervised sample episodes are collected to estimate the SAIL gradient, using plain stochastic gradient descent with a learning rate of $0.1$ for the temporal difference update and RMSprop with a a learning rate of $0.01$ for updating the policy. Figure 1b shows that SAIL stably converges to a policy that significantly outperforms the supervised baseline. While we do not expect SAIL to act optimally in previously unseen states but to instead exhibit recovery behavior, it is interesting to measure on how many states the learned policy agrees with the optimal policy using a soft count for each state based on the probability of the optimal action. Figure 1a shows that the amount of states in which the agent takes the optimal action roughly doubles its advantage over random chance and that the learned behavior is significantly closer to the optimal policy on states seen during execution.

### 4.1.2 Off-policy sample efficiency

For our second experiment, we evaluate the sample efficiency of SAIL by reusing previous sample episodes. As a temporal difference method, SAIL can be adapted using any off-policy temporal difference learning technique. In this work we elected to use truncated importance weights [16] with emphatic decay [24]. We evaluate the performance of SAIL collecting one new unsupervised sample episode in each iteration, reusing the samples collected in the past 19 episodes and compare the results against our implementation of Relative Entropy IRL[1]. We found that the importance sampling approach used by RelEntIRL makes interactions obtained by a pre-trained policy ineffective when using a tabular policy[1] and thus collect samples by taking actions uniformly at random. For comparability, we also evaluated SAIL using a fixed set of samples obtained by following a uniform policy. In this case, we found that the temporal-difference learning can become unstable in later iterations and thus decay the learning rate by a factor of $0.995$ after each iteration.

We vary the number of unsupervised sample episodes and show the score achieved by the trained policy in Figure 2. The score for RelEntIRL is measured by computing the optimal policy given the learned reward function. Note that this requires a model that is not normally available. We found that in this domain depending on the obtained samples, RelEntIRL has a tendency to learn shortcuts through the off-track area. Since small changes in the reward function can lead to large changes in the final policy, we average the results for RelEntIRL over 20 trials and bound the total score from

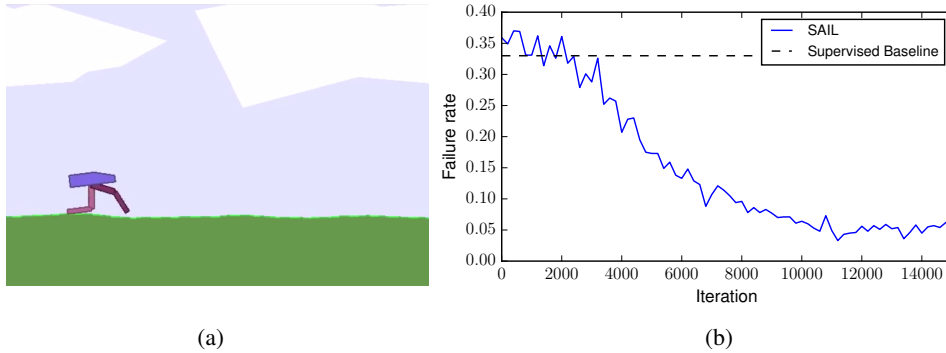

|     |     |
| :-: | :-: |
| (a) | (b) |

Figure 3: a) The bipedal walker has to traverse the plain, controlling the 4 noisy joint motors in its legs. b) Failure rate of SAIL over 1000 traversals compared to the supervised baseline measured. After 15000 iterations, SAIL traverses the plain far more reliably than the baseline.

below by the score achieved using the supervised baseline. We can see that SAIL is able to learn a near optimal policy using a low number of sample episodes. We can furthermore see that SAIL using uniform samples is able to learn a good policy and outperform the RelEntIRL baseline reliably.

## 4.2 Noisy bipedal walker

For our second experiment, we evaluate the performance of SAIL on a noisy variant of a two-dimensional Bipedal walker domain (see Figure 3a). The goal of this domain is to learn a policy that enables the simulated robot to traverse a plain without falling. The state space in this domain consists of 4 dimensions for velocity in x and y directions, angle of the hull, angular velocity, 8 dimensions for the position and velocity of the 4 joints in the legs, 2 dimensions that denote whether the leg has contact with the ground and 10 dimensions corresponding to lidar readings, telling the robot about its surroundings. The action space is 4 dimensional and consists of the torque that is to be applied to each of the 4 joints. To make the domain more challenging, we also apply additional noise to each of the torques. The noise is sampled from a normal distribution with standard deviation of 0.1 and is kept constant for five consecutive frames at a time. The noise thus has the ability to destabilize the walker. Our goal in this experiment is to learn a continuous policy from demonstrations, mapping the state to torques and enabling the robot to traverse the plain reliably. As a demonstration, we provide a single successful crossing of the plain. The demonstration has been collected from an oracle that has been trained on the bipedal walker domain without additional noise and is therefore not optimal and prone to failure. Our main metric for success on this domain is failure rate, i.e. the fraction of times that the robot is not able to traverse the plain due to falling to the ground. While the reward metric used in [2] is more comprehensive as it measures speed and control cost, it cannot be expected that a pure imitation learning approach can minimize control cost when trained with an imperfect demonstration that does not achieve this goal itself. Failure rate, on the other hand can always be minimized by aiming to reproduce a demonstration of a successful traversal as well as possible.

To represent our policy, we use a single shallow neural network with one hidden layer consisting of 100 nodes with tanh activation. We train this policy using a pure supervised approach as a baseline as well as with SAIL and contrast the results. During evaluation and supervised training, the output of the neural network is taken to be the exact torques whereas SAIL requires a probabilistic policy. Therefore we add additional Gaussian noise, kept constant for 8 consecutive frames at a time.

To train the network in a purely supervised approach, we use RMSProp over 3000 epochs with a batch size of 128 frames and a learning rate of $10^{-5}$. After the training process has converged, we found that the neural network trained with pure supervised learning fails 1650 times out of 5000 runs.

To train the policy with SAIL, we first initialize it with the aforementioned supervised approach. The training is then followed up with training using the combined gradient estimated by SAIL until the failure rate stops decreasing. To represent the gradient of the logarithmic stationary distribution, we use a fully connected neural network with two hidden layers of 80 nodes each using ReLU activations. Each episode is split into mini-batches of 16 frames. The $\nabla_\theta \log d^{\pi_\theta}$-network is trained using RMSProp with a learning rate of $10^{-4}$ whereas the policy network is trained using RMSprop

and a learning rate of $10^{-6}$, starting after the first 1000 episodes. As can be seen in Figure 3b, SAIL increases the success rate of 0.67 achieved by the baseline to 0.938 within 15000 iterations.

## 5   Conclusion

Imitation learning has long been a topic of active research. However, naive supervised learning has a tendency to lead the agent to states in which it cannot act with certainty and alternative approaches either make additional assumptions or, in the case of IRL methods, address this problem only indirectly. In this work, we proposed a novel imitation learning algorithm that directly addresses this issue and learns a policy without relying on intermediate representations. We showed that the algorithm can generalize well and provides stable learning progress in both, domains with a finite number of discrete states as well as domains with continuous state and action spaces. We believe that explicit reasoning over states can be helpful even in situations where reproducing the distributions of states will not result in a desirable policy and see this as a promising direction for future research.

### Acknowledgements

This work was supported by the Office of Naval Research under grant N000141410003

## Footnotes

[1]The original work by Boularias et al. shows that a pre-trained sample policy can be used effectively if a trajectory based representation is used

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
