[Supplementary Material]

# Appendix: State Aware Imitation Learning

## A  Convergence when using linear function approximation

We show convergence of SAIL when using linear function approximation to approximate $\nabla_\theta \log d^{\pi_\theta}(s)$. The proof is largely implied by the seminal findings of Tsitsiklis and Van Roy[25] analyzing convergence of average reward temporal difference learning. In this part we phrase the SAIL algorithm in their framework and show that the convergence results still apply. Given a vector of rewards for each state $g$, Tsitsiklis and Van Roy define the differential value vector for $J^* := \sum_{t=0}^\infty (g - \mu^* e)$ where $\mu^* = E_i[g(i)]$ is the average reward under the stationary state distribution and $e$ corresponds to the vector with all entries equal to 1. Learning a linear approximation $J_\omega = \omega \Phi$ with parameters $\omega$ and a matrix $\Phi$ with each row corresponding to the features $\phi(s)$ of each state $s$, they show that

1. The average cost temporal difference update of $\omega^{(k+1)} = \omega^{(k)} + \alpha\phi(s)\left(g(s) + \omega^{(k)}\phi(s_{+1}) - \omega^{(k)}\phi(s)\right)$ where $s_{+1}$ denotes the next state converges to $\omega^*$ where $\omega^*$ is the unique solution to $E[g(s) + \omega^{(k)}\phi(s_{+1}) - \omega^{(k)}\phi(s)|q(s)] = 0$ and $q(s)$ denotes the limiting distribution of the samples used to perform this update. (Theorem 2)

2. Defining the backup operator $TJ = g - \mu^* e + PJ$ where $P$ denotes the transition probability matrix of the MDP. If $q(s)$ is the stationary state distribution of the MDP, the unique solution corresponds to the projected fix point of the $\Phi\omega^* = \Pi T(\Phi\omega^*)$ where $\Pi$ denotes the projection operator performing the shortest projection onto the space that can be represented by linear combinations of the chosen features.

3. Any fix point $J$ of the backup operator $T$ is equal to $J^* + ce$ for an unknown constant $c$

We now show that SAIL performs average cost temporal difference learning for each dimension $i$ to estimate $\nabla_{\theta_i} \log \pi_\theta(a|s)$. We define the row corresponding to state $s$ in the cost vector $g$ to be $E[\nabla_{\theta_i} \log \pi_\theta(a_{-1}|s_{-1})|p(s_{-1}, a_{-1}|s)]$ and first make the observation that this implies $\mu^* = 0$, as

$$\mu_i^* = \int d^{\pi_\theta}(s) p(a_{-1}, s_{-1}|s) \nabla_{\theta_i} \log \pi_\theta(a_{-1}|s_{-1}) ds, a_{-1}, s_{-1}$$

$$= \int d^{\pi_\theta}(s_{-1}) \int \pi_\theta(a_{-1}|s_{-1}) \nabla_{\theta_i} \log \pi_\theta(a_{-1}|s_{-1}) da_{-1} ds_{-1} = \int d^{\pi_\theta}(s_{-1}) \cdot 0 ds_{-1} = 0.$$

Thus we observe that the SAIL update rule in the linear case

$$\omega_i^{(t+1)} = \omega_i^{(t)} + \alpha\phi(s)(\nabla_{\theta_i} \log \pi_\theta(a|s) + \omega_i\phi(s_{-1}) - \omega_i\phi(s))$$

corresponds to average reward temporal difference learning on the Markov chain induced by the reverse transition probabilities $p(s_{-1}|s, \pi_\theta)$ and Theorem 1 of [25] holds. Next, we note that the samples are observed by following the forward transitions and the limiting distribution of samples collected this way is equal to the stationary state distribution of Markov chain induced by the backwards transition dynamics as shown by Theorem 1 in [15]. As per the above, the update thus converges to a projected fix point $J_i \approx J_i^* + ce$ for each dimension $i$. We now note that as per definition and using [15], we have

$$J^* = \sum_{t=0}^\infty (g - \mu^* e) = \sum_{t=0}^\infty \nabla_\theta \log \pi_\theta(s_{-t}|a_{-t}) = \nabla_\theta \log d^{\pi_\theta}(s)$$

and thus
$$\nabla_\theta \log d^{\pi_\theta}(s) = J(s) - E[J(s)|d^\pi(s)].$$

This shows that the SAIL update rule combined with the subsequent adjustment for the mean converges to an estimate of the $\nabla_\theta \log d^{\pi_\theta}(s)$ in the case of linear function approximation.