[Reviews · NeurIPS 2017]

Reviewer 1



This paper proposes a framework for MAP-estimation based imitation learning, which can be seen as adding to the standard supervised learning loss a cost of deviating from the observed state distribution. The key idea for optimizing such a loss is a gradient expression for the stationary distribution of a given policy that can be estimated online using policy rollouts. This idea is an extension of a previous result by Morimura (2010). I find the approach original, and potentially interesting to the NIPS community. The consequences of deviating from the demonstrated states in imitation learning have been recognized earlier, but this paper proposes a novel approach to this problem. I have some problems with the presentation. The paper heavily builds on technical results from Morimura (2010). It would be helpful to present the work in a more self-contained format. See also the detailed comments below. The correctness of Eq. (3) and (4) depends on certain ergodicity assumptions of the MDP (required for detailed balance to hold), which are mentioned in Morimura (2010) but not in the current paper. Such assumptions should be added and discussed. The requirement of taking additional policy rollouts (in addition to the imitation learning data) is quite restrictive, even if these samples are unsupervised. In this respect, I wonder if a relevant comparison would be to take additional RL rollouts with a cost for ‘staying close’ to the demonstrated data. Such a cost might be hard to tune, and I would thus expect the current method to perform much better. Detailed comments: 158: This equation, which is fundamental to the work, is non-trivial, and uses ‘backward’ MDP transitions which are non-standard in RL/policy gradient. It would be helpful for the reader to provide more explanation, and explicitly refer to Morimura’s work for a proof. 172: Perhaps add some motivation for this constraint? 193: where is this assumption mentioned in this work? This whole paragraph can only be understood after reading the work of Morimura (2010). 205: How is this assumption justified? I have read the author feedback.

Reviewer 2



This paper presents a new approach to imitation learning called State-Aware Imitation Learning. Rather than just trying to match the actions of the demonstrations, it also tries to match the state distribution from the demonstrations. The paper adds matching that state distribution as part of the objective function and they derive a way to approximate the gradient for the state distribution. Results on two domains show that this approach is better able to overcome stochasticity in the environment than pure supervised learning. The idea of looking at the distribution over states seems very similar to the recent work on Generative Adversarial Imitation Learning (GAIL), where the discriminator network looks at the distribution over states or state-actions to classify whether the policy is from the demonstrations or the generating network. This work should be included in the related work and ideally compared to in the experiments. The paper says that sample episodes are collected in an "unsupervised" way multiple times throughout the paper. What exactly does this mean? They're collected by following the current policy of the algorithm, right? Both of the domains are setup in a particular way to benefit this algorithm: there is stochasticity in the environment to push the agent out of the state distribution of the demonstrations, and there is a reward penalty for leaving that distribution by too much (either going off track in the racing case, or the robot falling over). It would be interesting to see if this approach provides a benefit in other domains that are not so well tuned to its strengths. In both experiments, the authors train a supervised baseline first and then run their algorithm starting from that policy network. What happens if SAIL is trained from the start without the supervised training of the policy network first? I think this is a good contribution to the imitation learning literature, and the results look nice, but it should compare with GAIL, at least in the related work. I read your feedback, thanks for the clarifications and glad to see you'll add the GAIL comparison.

Reviewer 3



I've read the feedback, thanks for the clarifications. %%% This paper deals with imitation learning. It is proposed to maximize the posterior (policy parameters given the expert state-action pairs). The difference with a classic supervised learning approach is that here the fact that the policy acts on a dynamical system is taken into account, which adds the likelihood of states given policy (the states should be sampled according to the stationary distribution induced by the policy). This additional term is addressed through a temporal difference approach inspired by the work of Morimusa et al. [12], other terms are handled in the classic way (as in supervised learning). The proposed approach is experimented on two domains, a small on and a much larger one, and compared mainly to the supervised baseline (ignoring the dynamics). This paper proposes an interesting, sound and well motivated contribution to imitation learning. I found the paper rather clear, but I think it could be clarified for readers less familiar with the domain. For example: * in Eq. 3, the reverse probability appears. It would help to at least mention it in the text (it is not that classical) * the constraint (l.168, expected gradient null) comes simply from the fact that the probabilities should sum to one. It would help a lot to say it (instead of just giving the constraint) * the function f_w is multi-output (as many components as policy parameters), it could help to highlight this in the text * the off-policy approach should be made explicit (notably, as it is a reversed TD approach, due to the reversed probabilities behind, do the importance weights apply to the same terms?) Section 2 is rather complete, but some papers are missing. There is a line of work that tries to do imitation as a supervised approach while taking into account the dynamics (which is also what is done in this paper, the dynamics inducing a stationary distribution for each policy), for example : * "Learning from Demonstration Using MDP Induced Metrics" by Melo and Lopes (ECML 2010) * "Boosted and Reward-regularized Classification for Apprenticeship Learning" by Piot et al. (AAMAS 2014) The second one could be easily considered as a baseline (it is a supervised approach whit an additional regularization term taking into account the dynamics, so the used supervised approach in the paper could be easily adapted) Regarding the experiments: * it would have been interesting to vary the number of episodes collected from the expert * for the off-policy experiment, it would be good to compare also to (on-policy) SAIL, as a number of gathered trajectories * for the walker, giving the number of input/output neurons would also help (28 in/4 out for the policy, 28 in/3200 out for the gradient estimator? so a 28:80:80:3200 network?) This makes big output networks, it could be discussed. * still for the walker, what about off-policy learning? Regarding appendix A, the gradient of the log-policy plays the role of the reward. Usually, rewards are assumed to be uniformly bounded, which may not be the case here (eg, with a linear softmax policy based on a tabular representation, the probabilities could be arbitrary close to zero, even if ever strictly positive). Isn't this a problem for the proof?